# Ebola Virus Isolation Using Huh-7 Cells has Methodological Advantages and Similar Sensitivity to Isolation Using Other Cell Types and Suckling BALB/c Laboratory Mice

**DOI:** 10.3390/v11020161

**Published:** 2019-02-16

**Authors:** James Logue, Walter Vargas Licona, Timothy K. Cooper, Becky Reeder, Russel Byrum, Jing Qin, Nicole Deiuliis Murphy, Yu Cong, Amanda Bonilla, Jennifer Sword, Wade Weaver, Gregory Kocher, Gene G. Olinger, Peter B. Jahrling, Lisa E. Hensley, Richard S. Bennett

**Affiliations:** 1Integrated Research Facility, Division of Clinical Research, National Institute of Allergy and Infectious Diseases, National Institutes of Health, 8200 Research Plaza, Frederick, MD 27102, USA; james.logue@nih.gov (J.L.); wvargas@mriglobal.org (W.V.L.); timothy.cooper@nih.gov (T.K.C.); rebecca.reeder@nih.gov (B.R.); byrumr@niaid.nih.gov (R.B.); nicole.deiuliis@nih.gov (N.D.M.); yu.cong@nih.gov (Y.C.); amanda.bonilla@nih.gov (A.B.); jennifer.sword@nih.gov (J.S.); wade.weaver@nih.gov (W.W.); gregory.kocher@nih.gov (G.K.); golinger@MRIGLOBAL.ORG (G.G.O.); jahrlingp@niaid.nih.gov (P.B.J.); lisa.hensley@nih.gov (L.E.H.); 2Biostatistics Research Branch, National Institute of Allergy and Infectious Diseases, 5601 Fishers Lane, Rockville, MD 20852, USA; jingqin@niaid.nih.gov; 3Emerging Viral Pathogens Section, National Institute of Allergy and Infectious Diseases, National Institutes of Health, 8200 Research Plaza, Frederick, MD 21702, USA

**Keywords:** Mice, Vero, Huh7, Monocyte Derived Macrophages, Virus isolation, Semen, Breast milk, Ebola virus

## Abstract

Following the largest Ebola virus disease outbreak from 2013 to 2016, viral RNA has been detected in survivors from semen and breast milk long after disease recovery. However, as there have been few cases of sexual transmission, it is unclear whether every RNA positive fluid sample contains infectious virus. Virus isolation, typically using cell culture or animal models, can serve as a tool to determine the infectivity of patient samples. However, the sensitivity of these methods has not been assessed for the Ebola virus isolate, Makona. Described here is an efficiency comparison of Ebola virus Makona isolation using Vero E6, Huh-7, monocyte-derived macrophage cells, and suckling laboratory mice. Isolation sensitivity was similar in all methods tested. Laboratory mice and Huh-7 cells were less affected by toxicity from breast milk than Vero E6 and MDM cells. However, the advantages associated with isolation in Huh-7 cells over laboratory mice, including cost effectiveness, sample volume preservation, and a reduction in animal use, make Huh-7 cells the preferred substrate tested for Ebola virus Makona isolation.

## 1. Introduction

Following the largest Ebola virus disease (EVD) outbreak in history (2013–2016), the World Health Organization reported a total of 28,616 confirmed, probable, and suspected EVD cases in Western Africa with a fatality rate of 39.5% [1,2]. With over 17,000 EVD survivors, clinicians are now identifying long-term complications associated with EVD and recovery [1]. Patients have experienced delayed viral RNA clearance in semen, aqueous humor of the eye, CSF, and likely breast milk [3,4,5]. Ebola virus (EBOV) transmission from male survivors to sexual partners has also been reported, including one case 470 days after EVD recovery and discharge [6,7,8]. To reduce transmission following recovery, male survivors receive risk reduction counseling, including encouraging abstinence from sex or correct use of a condom during every interaction [9,10]. In Liberia, the Liberia’s Men’s Health Screening Program provided screening of semen from men following EVD recovery via reverse transcriptase-quantitative polymerase chain reaction (RT-qPCR) [10]. However, persistent RNA in human survivor samples was only identified by RT-qPCR screening which identifies samples containing viral genomic material that may or may not be infectious. Evidence of replicating EBOV has recently been shown from EVD survivor semen samples using SCID laboratory mice, Vero E6 cells, and HepG2 cells [11]. However, virus isolation from other sample types, such as breast milk, and isolation methods have not been reported. Virus isolation from these lower virus concentration samples, especially samples from later stages of EVD recovery, will be important to inform the public of health practices that are suggested from this men’s health program.

To ascertain whether a clinical sample is infectious, virus growth must be confirmed in vitro or in vivo. For filoviruses, the ability to detect and isolate a virus is reportedly 4-fold more sensitive using suckling laboratory mice than by using Vero E6 plaque assay [12]. However, only limited work has assessed isolation sensitivity for the new Ebola virus (EBOV) isolate, Makona (EBOV/Mak) [11]. Here we compare EBOV/Mak isolation sensitivity in Vero E6 (African green monkey [AGM kidney]), Huh-7 (human hepatocellular carcinoma), primary human monocyte-derived macrophages (MDMs), and in suckling BALB/cAnNCrl (BALB/c) laboratory mice. In the present study, initial EBOV/Mak mouse exposure did not result in lethal infection as previously reported for other EBOV isolates [13]. To confirm previous results, a comparison between EBOV/Mak and the Mayinga EBOV isolate (EBOV/Yam-May) was performed in vivo. Breast milk and semen matrix toxicity was also assessed using the isolation methods identified. In the present study, cell culture inoculation was as sensitive as laboratory mouse intracranial (IC) inoculation for virus isolation when using the same volume of inoculum. Though cell toxicity has been previously reported for semen in vitro, this study also identified that breast milk can cause toxicity, even at low concentrations [14,15]. This study was undertaken to identify virus isolation methods that will provide the greatest likelihood for isolation success prior to utilizing limited-volume human samples. The parameters tested in this study are not only applicable for EBOV but should be considered when developing isolation methods for other emerging viral infections.

## 2. Material and Methods

### 2.1. Cell Culture

Vero C1008 (Vero E6, [obtained through BEI Resources, NIAID, NIH: VERO C1008 (E6), Kidney (African green monkey), Working Cell Bank, NR-596]) and human hepatocellular carcinoma (HuH-7) cells obtained from Dr. Yoshimi Tsuda (Research Institute for Microbial Diseases, Osaka University) were cultured in Dulbecco’s modified Eagle’s medium with l-glutamine (DMEM, Lonza 12-604Q, Williamsport, PA, USA) supplemented with 10% heat-inactivated fetal bovine serum (HI-FBS, Gibco, ThermoFisher Scientific, Waltham, MA, USA) and incubated at 37 °C and 5% CO_2_. Human monocyte-derived macrophages (MDM) cells were cultured from human whole blood (medication-free) (100-03-04, Biological Specialty Corporation Colmar, PA, USA) and characterized as previously described [16].

### 2.2. Viruses

The C05 isolate of EBOV/Mak (full designation: Ebola virus/H.sapiens-tc/GIN/2014/Makona-C05, abbreviated name: EBOV/Mak-C05, GenBank accession number KP096420.1) was generously provided by Dr. Gary Kobinger of Public Health Agency Canada [17]. This stock was passaged three times in Vero E6 cells (obtained through BEI Resources, NIAID, NIH: VERO C1008 (E6), Kidney (African green monkey), Working Cell Bank, NR-596) in MEM-α, L-alanyl-L-glutamine without nucleosides (Gibco, ThermoFisher Scientific) supplemented with 2% US-origin, certified HI-FBS. All work with EBOV/Mak was performed in the Biosafety Level 4 laboratory at the Integrated Research Facility in Frederick, Maryland.

EBOV/Yam-May (full designation: Ebola virus H. sapiens-tc/COD/1976/Yambuko-Mayinga, abbreviated name: EBOV/Yam-May, GenBank accession number KY425656) was propagated from a 1:100 dilution (MOI = 0.0259) of the Master stock (BioSample SAMN0589700, IRF0164_EBOV) in VERO C1008 (NR-596, see above). Virus stock was propagated using MEM-α, L-alanyl-L-glutamine without nucleosides (Gibco, ThermoFisher Scientific) supplemented with 2% US-origin, certified, HI-FBS, (Gibco, ThermoFisher Scientific). Following harvest, HI-FBS was added to attain a 10% final concentration prior to cryopreservation. All work with EBOV/Yam-May was performed in the Biosafety Level 4 laboratory at the Integrated Research Facility in Frederick, Maryland.

### 2.3. Normal Breast Milk and Semen

Human breast milk (BM, single donor IR100042-18487, Innovative Research, Inc., Novi, MI, USA) and semen (single donor [single ejaculation] 991-04-SE, LEE Biosolutions, Maryland Heights, MO, USA) samples were commercially acquired.

### 2.4. Plaque Assay

Each test sample was serially diluted 1:10 for a total of 3 dilutions in α-MEM enriched with 10% HI-FBS and antibiotic-antimycotic solution (Gibco, ThermoFisher Scientific). The final concentration of penicillin, streptomycin, amphotericin B were 100 units/mL, 100 µg/mL, and 0.25 µg/mL, respectively. 300 µL from each sample dilution was then added to a 90% confluent monolayer of Vero E6 cells and incubated at 37 °C and 5% CO_2_ for 1 h with rocking every 10 ± 5 min. After incubating, a 1:1 ratio of 2.5% Avicel biopolymer and Temin’s modified Eagle medium (ThermoFisher Scientific) supplemented with 10% FBS, antibiotic-antimycotic solution, and L-alanyl-L-glutamine was overlaid and plates were incubated at 37 °C and 5% CO_2_ for 8 days. After the 8 days of incubation, the Avicel overlay was removed and a 0.2% crystal violet stain (Ricca Chemical, 3233-16, Pocomoke City, MD, USA) prepared in 10% neutral buffered formalin (Thermo Scientific, 5705) was added to each well and incubated at room temperature for 30 min. Cell monolayers were then washed with water, and plaques were counted manually.

### 2.5. TCID_50_ Assay

EBOV half maximal tissue culture infectious dose (TCID_50_) was titrated as previously described [18]. Briefly, matrix diluent was added to two control rows of each plate, and all test samples were added in quadruplicate and serially diluted 1:10 in a 96-well plate containing a 90% confluent Vero E6 monolayer. The outer wells of the plate were not used in the assay and remained full of cell culture media to avoid an edge effect. Following 14 days of incubation at 37 °C and 5% CO_2_, cytopathic effect was visually observed with a simple compound microscope. The TCID_50_ was quantified using the Reed-Muench method [19].

### 2.6. Quantitative Real Time RT-PCR

Total RNA was isolated as described previously [20]. Briefly, 70 µL of sample inactivated by Trizol LS was added to 280 µL of buffer AVL (Qiagen, Gemantown, MD, USA) with carrier RNA. Samples were then extracted using the QIAamp Viral RNA Mini Kit (Qiagen) in accordance with the manufacturer’s instructions, eluted in 70 µL of buffer AVE (Qiagen), aliquoted, and frozen. Viral RNA titer was determined using an experimental BEI Resources Critical Reagents Program EZ1 qRT-PCR kit assay in accordance with the manufacturer’s instructions. Sample titers were reported as RNA genome equivalents (GE) per mL of sample.

### 2.7. Sample Matrix Toxicity In Vitro

Vero E6, Huh-7, and MDM cells were grown in T25 flasks in DMEM with 10% HI-FBS. After 24 h of culture, flasks were confirmed to be 90% confluent and cell culture media was replaced with varying volumes (0.5 µL, 5 µL, 50 µL, or 100 µL) of normal (EBOV-negative) breast milk, semen, or supernatant from normal breast milk or semen that was centrifuged for 10 min at 10,000× *g* (clarified) in a volume adjusted to 1 mL with media. Following a 1 h incubation at 37 °C and 5% CO_2_, an additional 9 mL of media were added to the flasks. Images of the cell monolayers were taken prior to and 1 h following initial inoculation using a DM IL LED microscope (Leica Microsystems Inc., Buffalo Grove, IL, USA). The cells were then incubated for a total of 10 days at 37 °C and 5% CO_2_, and additional images were taken on days 3, 7, and 10 following inoculation. Day 10 images were captured after a single phosphate-buffered saline wash.

### 2.8. Sample Matrix Toxicity In Vivo

All animal experimental procedures were approved by the NIAID Division of Clinical Research (DCR), Animal Care and Use Committee and were performed in compliance with the Animal Welfare Act regulations, Public Health Service policy, and the Guide for the Care and Use of Laboratory Animals recommendations. The subjects were housed in an Association for Assessment and Accreditation of Laboratory Animal Care (AAALAC) International-accredited facility under BSL-4 conditions.

Litters of BALB/c laboratory mice (dam and suckling pups) (Charles River Laboratories strain code 028, Wilmington, MA, USA) were housed in individual isolator cages with food and water provided ad libitum. Two litters of up to six suckling laboratory mouse pups each were injected IC with 10 µL of normal (EBOV-negative) breast milk, semen, or cell culture media. All animals were observed twice daily for 10 days for clinical signs after which all animals were humanely euthanized.

### 2.9. EBOV/Mak Isolation In Vitro

Ten µL of each virus dilution (target dose 0.1 to 10,000 PFU/mL) in cell culture media was added to each well of a 6-well plate containing a 90% confluent monolayer of cells (Vero E6, Huh-7, or MDM) and 300 µL of the respective cell culture media. The plates were incubated for 1 h at 37 °C and 5% CO_2_ and rocked every 15 min. Following incubation, 1.7 mL of cell culture media were added to each well for a total of 2 mL per well and incubated for 10 days at 37 °C and 5% CO_2_. Following incubation, cell culture supernatant was collected and titered by plaque assay as described above.

The process was repeated with virus diluted in single donor semen, breast milk, or cell culture media and added to 6-well plates containing a 90% confluent monolayer of cells (Vero E6, Huh-7, or MDM) and 300 µL of the respective cell culture media. Following 10 days of incubation, virus isolation was confirmed by immunostaining p2 of each virus isolation attempt as previously described [16].

### 2.10. EBOV/Mak Isolation In Vivo

Laboratory mouse pups were inoculated IC with 10 µL of each virus dilution (target dose 0.1 to 10,000 PFU/mL) with one target dose/litter (total of 7–11 pups/dose). The animals were observed for 10 days post-inoculation for signs of disease including head tilt, circling, or paralysis, and any surviving animals were humanely euthanized. At necropsy, whole brain tissue was collected and frozen before processing. Pooled brain tissues from each dose group were thawed and added to a pre-weighed 50-mL closed ultra tissue grinder tubes (Fisher Scientific, 02-542-11, Pittsburgh, PA, USA), and phosphate-buffered saline containing 10% HI-FBS was added at a 1:10 *w*:*v* ratio before homogenizing. The brain homogenate was clarified by low speed centrifugation (3000 revolutions per minute for 10 min at 4 °C) and aliquoted, and samples were frozen at −80 °C prior to virus titration by plaque assay as described above. Animals that died within the first 48 hours following exposure were excluded from analysis.

### 2.11. In Vivo Virus Isolation Comparison between Two EBOV Isolates

To confirm our initially collected EBOV/Mak lethality data in suckling BALB/c laboratory mice and previously reported data comparing EBOV/Mak and EBOV/Yam-May lethality in laboratory mice [21], the virus isolation as described previously was completed using either EBOV/Mak or EBOV/Yam-May isolates in two litters of pups/target dose. Following exposure, suckling laboratory mice were observed for 28 days post-exposure. For laboratory mice that succumbed during days 3–27 post-exposure, whole brain tissue was collected and frozen before virus titer was determined as described above. Brain tissue was individually harvested from laboratory mice that survived to the end of the study (day 28 post-exposure) and divided in half for virus titration (plaque assay and RT-qPCR) and histology. Animals that died within the first 48 h following exposure were excluded from analysis.

### 2.12. Histology/Immunohistochemistry

Histology and immunohistochemistry were performed only on brain tissues collected from animals that survived to 28 days post-exposure. Brains were fixed for 72 h in 10% neutral buffered formalin before automated processing in a Tissue-Tek VIP-6 vacuum infiltration processor (Sakura Finetek USA, Torrance, CA, USA) followed by paraffin embedding. Tissues were sectioned by a rotary microtome (model 2255, Leica Biosystems, Buffalo Grove, IL, USA) at a 4-µm thickness, stained with hematoxylin and eosin (H&E) and cover slipped, and read by a veterinary pathologist blinded to treatment.

To detect EBOV RNA in formalin-fixed, paraffin-embedded (FFPE) tissues, in situ hybridization (ISH) was performed using the RNAscope 2.5 HD RED kit (Advanced Cell Diagnostics, Newark, CA, USA) according to the manufacturer’s instructions. Briefly, 20 ZZ probe pairs targeting the EBOV *VP40* gene were designed and synthesized by Advanced Cell Diagnostics (catalogue 300031). After deparaffinization with xylene, a series of ethanol washes and peroxidase blocking, sections were heated in Target Retrieval Buffer (Advanced Cell Diagnostics) and then digested by proteinase 5-chloro-2-methyl-3(2H)-isothiazolone with 2-methyl-3(2H)-isothiazolone. Sections were exposed to ISH target probe and incubated at 40 °C in an oven hybridization system (HybEZ II, Advanced Cell Diagnostics) for 2 h. After rinsing, the ISH signal was amplified using company-provided Pre-amplifier and Amplifier conjugated to horseradish peroxidase) and incubated with chromogenic-RED substrate (composed of reagents Fast RED-A and Fast RED-B, Advanced Cell Diagnostics) for 10 min at room temperature. Sections were then counterstained with hematoxylin, dehydrated and coverslipped.

## 3. Results

### 3.1. EBOV/Mak In Vitro Isolation Comparison

To identify virus titers that fall above and below the titer at which virus can be isolated in vitro, cell culture-grown EBOV/Mak was serial diluted to a target titer from 0.1 to 10,000 PFU/mL, and three aliquots per dilution were frozen. Samples were thawed, and titers were quantified by traditional plaque assay, TCID_50_ assay, and qPCR (Table 1), and in vitro virus isolation attempts were completed on these samples in parallel (Table 2). Virus was detected by the three quantification methods in each of the three experiments at the dilution containing a target dose of 10 PFU/mL. However, this dilution is below the limit of quantification for the plaque assay and may not accurately represent an actual sample titer.

In vitro virus isolation was attempted in parallel using Vero E6, Huh-7, and MDM cell lines. A total of 10 µL from each sample was added to each well and incubated for 10 days before virus isolation was confirmed by plaque assay. The sample volume used for these assays corresponds to the maximum volume injected IC in suckling laboratory mice, and the number of wells chosen (*n* = 6) corresponded roughly to the anticipated number of laboratory mice per litter. Virus was isolated in all three cell lines at least once at the target PFU/mL of 10, and only once from samples containing a target titer of 0.1–1 PFU/mL in Huh-7 cells (Table 2). Virus titers for all successful in vitro isolation attempts were greater than 3.3 × 10^4^ EBOV/Mak titers were determined by plaque assay (Table 2).

EBOV/Mak isolation in suckling laboratory mice was completed in parallel in three cell types using a single virus sample preparation across methods (isolation attempt 4, Table 2). By day 10 post-exposure, virus amplification had occurred in the absence of clinical signs of disease, and virus titer had reached 2.08 × 10^3^ to 3.84 × 10^4^ PFU/g brain tissue. Inoculation of suckling mice was repeated with observations extending through day 28 (isolation attempt 5, Table 2).

### 3.2. In Vivo Virus Isolation Comparison between Two Ebola Virus Isolates

Because IC exposure to EBOV/Mak did not result in a lethal phenotype in suckling mice as expected, sucking mice were exposed IC to serial dilutions of EBOV/Mak and EBOV/Yam-May (a known lethal virus) in parallel and observed for 28 days. At the end of the 28-day observation, survival curves were generated for each target dose group for both EBOV isolates (Figure 1). Fatality in the first 48 h after inoculation was not included in the percent survival calculation or brain tissue tested for EBOV since death was likely due to complications of injection and not EBOV disease progression. The one death in the EBOV/Mak negative control (media) group (*n* = 2) was not associated with viral disease.

Fifty percent lethal dose (LD50) for both EBOV/Mak and EBOV/Yam-May were calculated using a logistic regression model and likelihood ratio statistic using a program developed in R. The estimated LD50 for EBOV/Mak is 4.56 PFU (95% confidence interval is 4.19 to 4.77 PFU). The estimated LD50 for EBOV/Yam-May is 6.46 PFU (95% confidence interval 4.19 to 6.68 PFU). No statistical difference in lethality was measured between the two isolates in this mouse model (likelihood ratio test, *P* = 1.0).

Virus titers for 10% brain homogenates were determined by plaque assay individually for laboratory mice that survived to 28 days post-inoculation or were grouped by target challenge dose if animals succumbed to disease. For animals that succumbed to disease, virus titer had reached 2.4 × 10^4^ to 1.94 × 10^6^ and 5.44 × 10^3^ to 4.12 × 10^5^ PFU/g brain tissue for the EBOV/Mak and EBOV/Yam-May groups respectively.

Individual brain homogenates collected from EBOV/Mak- or EBOV/Yam-May-exposed laboratory mice surviving to the end of the study (day 28 post-inoculation) were negative for EBOV by plaque assay; however, viral RNA persistence was measured by plaque assay. Clinical signs of disease (e.g., scruffy fur, head tilt, circling) were observed in some animals and did not resolve by 28 days post-inoculation. These signs were not present in animals inoculated with cell culture media only.

### 3.3. EBOV/Mak Isolation Attempt Summary

EBOV/Mak isolation attempts were completed with aliquots of diluted virus using the same target input virus titer and volume for all in vitro and in vivo methods. Mouse isolation attempts (2 total) were completed in parallel with in vitro isolation attempts 4 and 5. When target virus titers ranged from 100 to 10,000 PFU/mL, all tested methods successfully isolated virus (100%). When target titers ranged from 1 to 10 PFU/mL, isolation success decreased to 25% (2/8) in Vero E6 cells, 50% (5/10) in Huh-7 cells, 38% (3/8) in MDM cells, and 50% (2/4) in mice. Isolation attempts were unsuccessful for all methods tested when the titer dropped below 1 PFU/mL (0%). At roughly the lowest titer of virus isolation success (10 PFU/mL), the slight variation in isolation outcomes were not statistically significant across all isolation methods (Fischer’s exact test, *P* > 0.05).

### 3.4. Histology and Detection of EBOV RNA in Suckling Laboratory Mice Brains

A number of brain lesions were present in IC-inoculated laboratory mice, ranging from minimal to severe. At the extreme end of the spectrum, diffuse and severe spongiosis and demyelination of the white matter of the cerebral cortex were noted in several laboratory mice (Figure 2). Variable dilation of the lateral ventricles (primary hydrocephalus versus hydrocephalus *ex vacuo*) was rarely accompanied by minimal to mild non-suppurative encephalitis. In the EBOV/Yam-May-inoculated laboratory mice, this lesion was present in two laboratory mice in the target dose 10 PFU/laboratory mouse group only. In the EBOV/Mak-inoculated laboratory mice, this lesion was present in in one laboratory mouse in the 10 PFU/laboratory mouse target dose group and two laboratory mice in the 0.1 PFU/laboratory mouse target dose group only.

A second distinct lesion, occasionally present in laboratory mice with white matter spongiosis, was the presence of few scattered to multiple coalescing nodular rests of small round blue cells superficial to the molecular layer of the cerebellum (external granular cell layer remnants, Figure 3). A variable minimal to low number of granule cells were present within the molecular layer. In the EBOV/Yam-May-inoculated laboratory mice, this lesion was present only in one laboratory mouse in the 10 PFU/laboratory mouse target dose group, three laboratory mice in the 1 PFU/laboratory mouse target dose group, and two laboratory mice in the 0.01 PFU/laboratory mouse target dose group. In the EBOV/Mak inoculated laboratory mice, this lesion was present only in two laboratory mice in the 10 PFU/laboratory mouse target dose group and one laboratory mouse in the 1 PFU/laboratory mouse target dose group.

In several laboratory mice, minimal to focally mild non-suppurative inflammation was present in the neural parenchyma, ranging from macrophages and lymphocytes expanding the perivascular Virchow-Robbins spaces and infiltrating the neuropil to discrete (micro)glial nodules (Figure 4). In the EBOV/Yam-May-inoculated laboratory mice, this lesion was present only in one laboratory mouse in the 10 PFU/laboratory mouse target dose group. In the EBOV/Mak-inoculated laboratory mice, this lesion was present only in one laboratory mouse each in the 1 PFU/laboratory mouse target dose group and 0.1 PFU/laboratory mouse target dose groups.

### 3.5. Breast Milk and Semen Induced Cytotoxicity Using In Vitro and In Vivo Isolation Methods

The matrix effect (cytotoxicity) of normal breast milk and semen was tested in MDM, Vero E6, and Huh-7 cells. Breast milk or semen, whether whole or clarified by centrifugation, were added to 90% confluent monolayers of each of the three cell types. A control flask with only media was incubated in parallel. Cell monolayers were photographed at days 0 (after 1-h incubation), 3, 7, and 10 post-exposure (not all images shown, Figure 5). Breast milk was cytotoxic on Vero E6 and MDM cells at volumes of 50 and 100 µL. However, breast milk visually appears less toxic in Huh-7 cells. Toxicity was initially identified (day 0 post-exposure) in the 100-µL treatment, and the cell monolayer recovered by day 10 post-exposure. Breast milk that was clarified by centrifugation was not toxic at the concentrations tested. Whole semen and semen clarified by centrifugation were both slightly cytotoxic in MDM cells but not Vero E6 or Huh-7 cells (Table 3). Images of cell monolayers at ten days following semen or breast milk exposure are displayed (Figure 5).

Additionally, two litters of suckling laboratory mice were injected IC with 10 µL of breast milk, semen, or cell culture media. No laboratory mice experienced fatality during the 10 days after injection (Table 3). Two animals exposed to semen did not recover from anesthesia and were not counted against the number of survivors.

To determine if matrix effect impacts virus isolation success, spiked semen and breast milk were inoculated onto VeroE6, Huh-7, or MDM cells as described above. Results were compared to media only preparations using the log binomial likelihood ratio statistic and compared to the chi-squared reference distribution with significance considered at the 0.05 level. Again, there was no statistical difference in isolation success for spiked media samples across cell types. VeroE6 and Huh-7 cells were both successful at isolating virus at the same dilutions (*p* = 0.96) and both demonstrated significantly better virus isolation success from spiked semen samples than MDM cells (*p* < 0.01). Virus was not isolated from any of the spiked breast milk samples even though the cell monolayers appeared healthy. Isolation success from spiked semen or media were not significantly different by across the cell types (Table 4).

## 4. Discussion

Following the largest recorded EVD outbreak in history (2013–2016), virus persistence in semen and breast milk was identified as a possible source of transmission after patients had recovered from clinical symptoms. Although viral RNA has been detected in these body fluids, limited work has assessed these fluids for infectious virus [11]. Thus, the true public health impact of RNA-positive fluids remains unknown. The present study optimized virus isolation methods both in vitro and in vivo and compared virus isolation success across methods. In addition, we assessed the in vitro and in vivo toxicity associated with non-traditional fluid samples, semen and breast milk. These efforts were undertaken to identify the best isolation method prior to attempting virus isolation from human samples, which are often limited in volume and availability.

Virus isolation was attempted from serial dilutions of cell-culture-grown EBOV to identify the lowest viral dilution from which virus could be successfully isolated using four isolation methods. Three cell types, all previously used in EBOV research, were selected for this comparison. Two immortalized cell lines, Vero E6 cells (African Green Monkey kidney cells) and Huh-7 cells (human liver cells), were selected for their availability, ease of culture, and known permissiveness to EBOV infection. We also included a primary human cell line from four donors, MDMs, which are thought to support EBOV replication during human infection. Additionally, IC inoculation of suckling laboratory mice was chosen as it is a common virus isolation method for a variety of viruses and was reported to be more sensitive than cell culture. EBOV/Mak was successfully isolated from samples containing a target dose of 10 PFU/mL in all three cell types and suckling BALB/c laboratory mice with no significant differences in sensitivity across methods.

Potential semen- or breast milk-associated matrix toxicity was also assessed for these same methods. Huh-7 cells and brains of suckling BALB/c laboratory mice were the most resistant to breast milk-associated matrix toxicity. Interestingly, concentrations of breast milk that were shown to be toxic when unaltered were not toxic following clarification by centrifugation. This suggests the matrix cytotoxicity of breast milk is most likely related to components of the sample that were centrifuged out. Whole semen and semen clarified by centrifugation were both slightly cytotoxic in MDM cells but not Vero E6 or Huh-7 cells, suggesting the cytotoxicity is related to soluble matrix components that were not removed by the centrifugation process. Though the normal semen and breast milk used in this study were from individual healthy donors, it is unknown how much matrix toxicity would vary between donors or health status. Virus isolation was not successful from spiked breast milk and the mechanism of virus replication inhibition is unknown. The isolation attempts were made by two different users on two separate days in parallel with the isolation spiked semen or media control. If virus isolation from breast milk samples will be pursued in the future, alternative isolation methods will need to be identified. Additionally, although components of semen have been reported to increase EBOV infection in HeLa cells, in the present study virus isolation from spiked whole semen was as sensitive as the media matrix samples. The difference in results from these two studies may be explained by the use of whole semen instead of seminal plasma or components of semen as well as a differences in the in vitro effects of these matrices on the cell lines used [22].

During the 10 day in vivo isolation comparison described above, disease fatality was not observed in BALB/c laboratory mice during the first 10 days post-exposure (Table 2). As fatality is normally used as an indicator of severe viral infection, additional tests were needed to establish how this virus isolate affected suckling BALB/c laboratory mice. Smither et al reported EBOV/Mak (C07 isolate) was associated with reduced fatality in A129 interferon α/β receptor-deficient laboratory mice compared to that observed in laboratory mice following exposure to EBOV H. sapiens-tc/COD/1976/Yambuko-Ecran (EBOV E718 isolate) [23]. To determine if the EBOV/Mak isolate also has an attenuated phenotype in BABL/c laboratory mice, we inoculated suckling BALB/c laboratory mice with either EBOV/Mak or EBOV/Yam-May (isolated from the same outbreak as EBOV E718) and observed animals for 28 days for signs of virus infection. Both virus isolates did cause lethal disease in suckling laboratory mice with no significant differences in LD50 and limited fatality observed prior to 10 days post-exposure, suggesting animals in our initial isolation comparison study may have been euthanized just prior to succumbing to disease. No significant differences in LD50 were measured between the two virus isolates tested. Though no virus was isolated from any animals surviving to 28 days post-exposure, signs of previous infection were apparent in some surviving animals including viral RNA persistence in the brain.

Given the matrix toxicity and sensitivity findings described above, virus isolations using either Huh-7 cells or suckling BALB/c laboratory mice were identified as the preferred EBOV/Mak isolation methods. However, in vitro virus isolation offers multiple potential advantages over the in vivo method. First, larger sample volumes can be tested in a single system in vitro than the limited IC injection volume (10 µL) used in the suckling BALB/c laboratory mouse model. While the present in vitro study used 10 µL of sample in a 6-well plate for comparison purposes to the in vivo method, this process could be scaled up to a T25 or T125 flask using larger sample volumes in a single system. Second, premature animal death and the resultant loss of sample is a potential major drawback to any in vivo method. In the present study, suckling BALB/c laboratory mouse fatality was observed in the first 48 h after inoculation that was not likely associated with virus exposure as no signs of disease were observed. Early fatality can waste sample prior to virus amplification which can be especially problematic when sample volume is limited. Isolating virus in a species not normally infected with the virus may raise regulatory concerns about adaption to a novel host. Additionally, with the Huh-7 isolation method, virus was isolated in the presence of antibiotics and antimycotics. Bacterial or fungal contamination from clinical samples may decrease the ability to isolate virus in vivo. Virus growth in Huh-7 cells also shows reduced to no mutation frequency as compared to virus growth in Vero E6 cells [24,25]. Reduced mutation frequency will be especially important if further characterization of virus found in breast milk and semen is to be completed. Finally, ethical considerations related to reducing the number of animals used in research and refinement of techniques to minimize animal use also favors in vitro isolation.

Any future isolation work should take note of a few caveats involved in this study. Limited clinical sample availability meant that we had to use an EBOV/Mak viral stock grown and passaged a total of three times in Vero E6 cells for isolation comparisons. Virus growth using Vero E6 cells can result in the replacement of a threonine with an isoleucine at positions 544 (T544I) in the EBOV glycoprotein and is present in the virus isolate we used for our comparison studies [25]. This mutation has been shown to increase EBOV-like-particle viral entry when tested in monkey kidney cells (COS-7; *p* < 0.01) and has been shown to increase EBOV replication in Vero E6 cells (*p* < 0.001) but no statistical difference was observed for Huh-7 cells. No statistical difference was reported for any other mutation tested, including the presence of a valine or adenosine at position 82 of the EBOV glycoprotein [26]. Additionally, while virus isolation from spiked breast milk and semen were attempted, the actual success rate for these samples may vary based on the test patient. The lower limits of isolation success may vary when testing clinical samples that may have varying degrees of protein or other biological contaminants. In the present study, the Huh-7 method has demonstrated similar isolation sensitivity to the other methods tested as well as multiple advantages, including scalability, isolation in the presence of antibiotics and antimycotics, resistance to matrix toxicity, and potentially reduced EBOV mutation frequency. Parameters considered in this work should be considered when identifying optimal isolation systems for other emerging and re-emerging viruses.

## Figures and Tables

**Figure 1 viruses-11-00161-f001:**
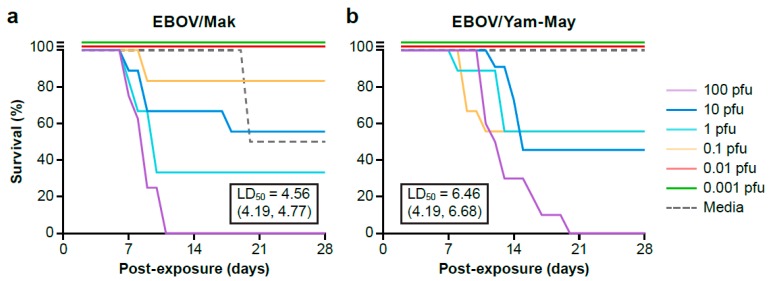
Laboratory mouse survival curves for the extended (**a**) EBOV/Mak and (**b**) EBOV/Yam-May cohorts. Laboratory mice are grouped by the target challenge dose. Fifty percent lethal dose (LD50) is indicated in the text box with 95% confidence interval in parenthesis.

**Figure 2 viruses-11-00161-f002:**
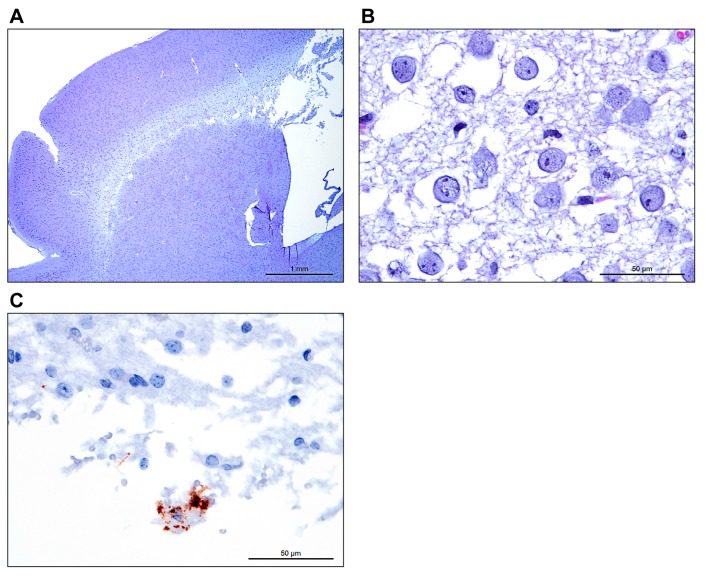
(**a**) Low power image taken at 25× total magnification, 2.5× objective shows diffuse rarefaction of the cerebral cortical white matter of the brain. (**b**) Higher magnification image taken at 600× total magnification, 60× objective shows abundant edema expanding the white matter with loss of myelin. (**c**) Using In situ hybridization targeting the EBOV genome, a large foamy macrophage (Gitter cell) is stained within the spongiotic white matter.

**Figure 3 viruses-11-00161-f003:**
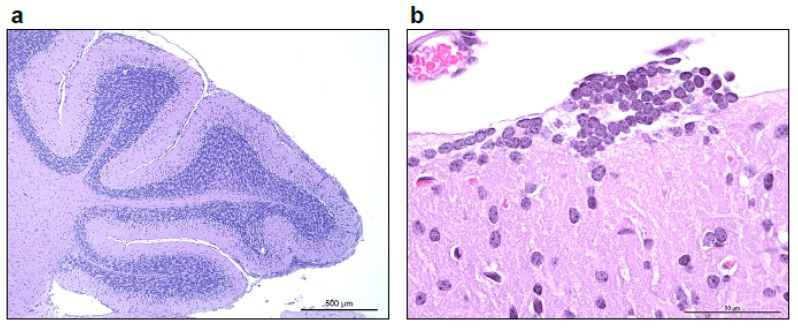
(**a**) Low power image taken at 50× total magnification, 5× objective shows multifocal to coalescing nodular remnants of the external granular layer of the cerebellum, with increased cellularity of the molecular layer. (**b**) Higher magnification image taken at 600× total magnification, 60× objective shows a rest of external granule cells superficial to the molecular layer.

**Figure 4 viruses-11-00161-f004:**
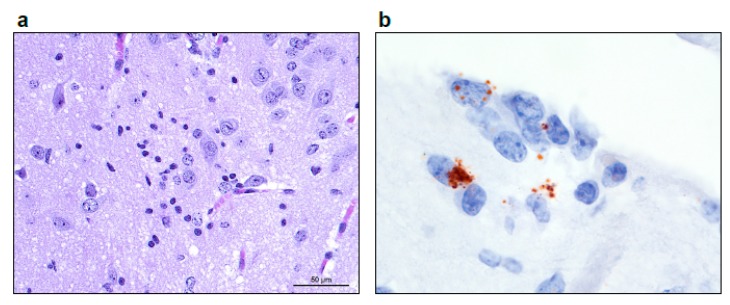
(**a**) Image taken at 400× total magnification, 40× objective of the (micro)glial nodule in the thalamus. (**b**) Using In situ hybridization targeting the EBOV genome, a glial nodule in the cortical white matter is stained and magnified at 1000× total magnification (oil), 100× objective.

**Figure 5 viruses-11-00161-f005:**
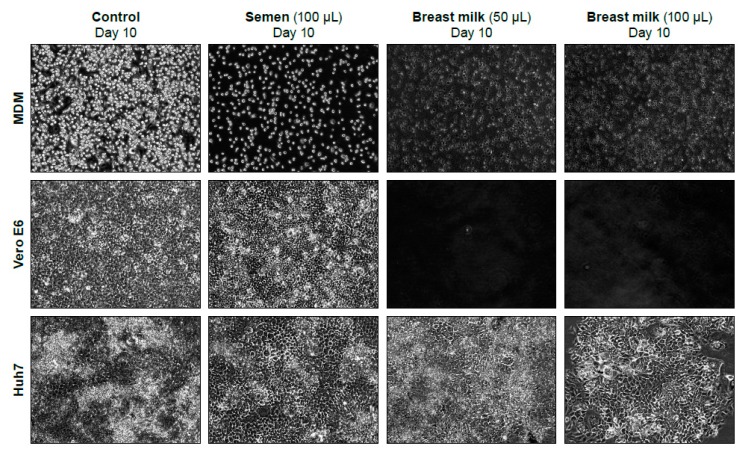
10× images of MDM, Vero E6, and Huh-7 cells taken 10 days following exposure to select volumes of normal semen, breast milk, or cell culture media (control). Breast milk was toxic at 50 and 100 µL/T-25 flask on Vero E6 cells (no cells present after 10 days) and MDM cells, but toxicity was less on these cells than on Huh7 cells.

**Table 1 viruses-11-00161-t001:** Detection of EBOV/Mak isolate from sample dilutions using three titer methods.

Target Titer (PFU/mL)	Titer by Method								
	PFU/mL			TCID_50_/mL			qPCR (GE/mL)		
	1	2	3	1	2	3	1	2	3
10,000	2400	7110	3590	2693	3953	5801	4.8 × 10^6^	4.6 × 10^6^	4.0 × 10^6^
1000	326	644	544	269	580	269	4.6 × 10^5^	5.8 × 10^5^	4.9 × 10^5^
100	43	76	48	58	27	6	3.8 × 10^4^	4.6 × 10^4^	5.4 × 10^4^
10	3 ^a^	4 ^a^	6 ^a^	6	2.7	2.7	5.5 × 10^2^	4.5 × 10^2^	5.0 × 10^3^
1	U	U	1 ^a^	U	2.7	U	U	U	U
0.1	U	U	U	U	U	U	U	U	U

^a^ Plaque assay titers listed are below the limits of quantification. Abbreviations: U: undetected.

**Table 2 viruses-11-00161-t002:** Summary of EBOV/Mak isolation success using three cell lines and suckling BALB/c laboratory mice.

Target PFU/Sample	EBOV/Mak Virus Isolation by Method														
	Vero E6				Huh-7					MDM				Mice	
	1	2	3	4	1	2	3	4	5	1	2	3	4	4 ^b^	5 ^c^
100	+	+	+	+	+	+	+	+	+	+	+	+	+	+	+
10	+	+	+	+	+	+	+	+	+	+	+	+	+	+	+
1	+	+	+	+	+	+	+	+	+	+	+	+	+	+	+
0.1 ^d^	-	+	-	+	+	+	- ^a^	+	+	-	+	+	+	+	+
0.01 ^d^	-	-	-	-	-	-	+ ^a^	-	-	-	-	-	-	-	-
0.001 ^d^	-	-	-	-	-	-	-	-	-	-	-	-	-	-	-

^a^ These samples were retested and results were confirmed. ^b^ Brain tissue harvested at day 10 from all animals. ^c^ Brain tissue harvested when animal succumbed to disease or on day 28 post-exposure (scheduled necropsy). ^d^ Plaque assay titers for these samples are below the lower limit of quantification. Abbreviations: “+” indicates successfully virus isolation attempt; “-” indicates unsuccessful virus isolation attempt.

**Table 3 viruses-11-00161-t003:** Presence or absence (“+” or “-“) of semen- or breast milk-associated matrix toxicity at varying concentrations in vitro and survival in vivo at day 10 post-exposure.

Sample	Cell Type	Sample Volume (µL)				Total Laboratory Mouse Survivors ^a^/Group
		0.5	5	50	100	
Breast milk	Vero E6	-	-	+ ^b^	+ ^b^	11/11
	Huh-7	-	-	-	+/- ^c^	
	MDM	-	-	+	+	
Breast milk clarified ^d^	Vero E6	-	-	-	-	ND
	Huh-7	-	-	-	-	
	MDM	-	-	-	-	
Semen	Vero E6	-	-	-	-	9/9
	Huh-7	-	-	-	-	
	MDM	± ^e^	± ^e^	± ^e^	± ^e^	
Semen clarified ^d^	Vero E6	-	-	-	-	ND
	Huh-7	-	-	-	-	
	MDM	± ^e^	± ^e^	± ^e^	+/- ^e^	
Media	Vero E6	-	-	-	-	12/12
	Huh-7	-	-	-	-	
	MDM	-	-	-	-	

^a^ Laboratory mouse total only includes animals that survived inoculation. IC inoculation volume was 10 µL. ^b^ Complete monolayer destruction. ^c^ Cell detachment occurred initially after incubation, and the monolayer recovered by day 10 post-exposure. ^d^ Samples were clarified at 10,000 g for 10 min at 4 °C. ^e^ Monolayers exposed to semen retained roughly 50% of the monolayer compared to that of the negative control monolayer. However, the remaining cells appear healthy. Abbreviations: “+” indicates presence of matrix toxicity; “-” indicates absence of matrix toxicity; “±” indicates initial matrix toxicity with cell layer recovery following incubation.

**Table 4 viruses-11-00161-t004:** Virus isolation success from spiked breast milk and semen samples (*n* = 12).

Target PFU/Sample	Virus Isolation Success in Spiked Samples by Cell Type ^a^ (Total Positive/12 Tested)								
	Semen			Breast Milk			Media		
	VeroE6	Huh-7	MDM	VeroE6	Huh-7	MDM	VeroE6	Huh-7	MDM
100	12	12	12	0	0	0	12	12	10
10	12	12	10	0	0	0	12	11	11
1	12	11	5	0	0	0	12	11	8
0.1	7	8	0	0	0	0	2	3	1
0.01	0	0	0	0	0	0	2	0	1
0.001	0	0	0	0	0	0	0	0	0

^a^ Isolation attempt completed in individual wells of 6 well plate (9 cm^2^/well). Sample volume added was 10 µL sample/well.

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
