# Peer review of "Ebola Virus Isolation Using Huh-7 Cells has Methodological Advantages and Similar Sensitivity to Isolation Using Other Cell Types and Suckling BALB/c Laboratory Mice"

_viruses, 2019, doi:10.3390/v11020161_

Round 1

Reviewer 1 Report

The authors have adequately addressed most of my questions. However, some minor points remain open:

"However, it is unclear if these fluids contain infectious virus." It seems clear that survivors of EVD can transmit the virus via sex. Therefore, the sentence should be revised.

"This study also identified that non-traditional sample matrices, such as breast milk or semen, can cause toxicity in vitro even at low concentrations." Toxicity of semen has previously been documented, see PMID: 20573198 for an example.

"Isolation success from spiked semen or media were not significantly different by across the cell types (Table7)." These results contradict those reported in PMID: 29941593 and reasons for these discrepant observations should be discussed.

The numbering of the tables must be updated.

Author Response

The authors have adequately addressed most of my questions. However, some minor points remain open:

"However, it is unclear if these fluids contain infectious virus." It seems clear that survivors of EVD can transmit the virus via sex. Therefore, the sentence should be revised. 

Sentence modified to “However, as there have been few cases of sexual transmission, it is unclear whether every RNA positive fluid sample contains infectious virus.” (Lines 26-28)

"This study also identified that non-traditional sample matrices, such as breast milk or semen, can cause toxicity in vitro even at low concentrations." Toxicity of semen has previously been documented, see PMID: 20573198 for an example.

Sentence modified to “Though cell toxicity has been previously reported for semen in vitro, this study also identified that breast milk can cause toxicity, even at low concentrations (Kim et al., 2010; Munch et al., 2007).” (Lines 72-74)

"Isolation success from spiked semen or media were not significantly different by across the cell types (Table7)." These results contradict those reported in PMID: 29941593 and reasons for these discrepant observations should be discussed. The numbering of the tables must be updated.

We appreciate the study the reviewer provided.  We have added the following to the discussion to address this concern:  “Although components of semen have been reported to increase EBOV infection in HeLa cells, in the present study virus isolation from spiked whole semen was as sensitive as the media matrix samples. The difference in results from these two studies may be explained by the use of whole semen instead of seminal plasma or components of semen as well as a differences in the in vitro effects of these matrices on the cell lines used (Bart, et al., 2018).”  (Lines 451-456)

The table numbers have been updated.

Reviewer 2 Report

Ebola virus isolation using Huh-7 cells has methodological advantages and similar sensitivity to isolation using other cell types and suckling BALB/c laboratory mice

In this manuscript, Logue et al evaluated the efficacy of isolating Ebola virus Makona (EBOV/Mak) from three cell lines Vero E6, Huh-7, and monocyte-derived macrophages, and suckling Balb/C laboratory mice. The authors compared the ability to isolate spiked virus from the alternative matrices like semen and breast milk using in vitro or in vivo approach.

The multiple major comments were addressed in the revised version of the manuscript. The presentation of the material coherent and is well-supported by conducted experiments and presented results. 

Minor suggestions:

The introduction and discussion about virus isolation in the text are focused on the Ebola/Makona isolate. However, in some tests, the Ebola/Mayinga virus was used for comparison with Ebola /Makona – see Materials and Methods (2.11),..”. The clarity of the story would benefit from the little bit more detailed explanation of why the Ebola/Mayinga isolate was chosen for comparison. 

Author Response

Minor suggestions:

The introduction and discussion about virus isolation in the text are focused on the Ebola/Makona isolate. However, in some tests, the Ebola/Mayinga virus was used for comparison with Ebola /Makona – see Materials and Methods (2.11),..”. The clarity of the story would benefit from the little bit more detailed explanation of why the Ebola/Mayinga isolate was chosen for comparison.

We’ve modified the following lines to clarify why EBOV Mayinga was added to the study:

“In the present study, initial EBOV/Mak mouse exposure did not result in lethal infection as previously reported for other EBOV isolates [13]. To confirm previous results, a comparison between EBOV/Mak and the Mayinga EBOV isolate (EBOV/Yam-May) was performed in vivo.” (Lines 67-70)

“Because IC exposure to EBOV/Mak did not result in a lethal phenotype in suckling mice as expected, sucking mice were exposed IC to serial dilutions of EBOV/Mak and EBOV/Yam-May (a known lethal virus) in parallel and observed for a full 28 days.” (Lines 265-267)

Reviewer 3 Report

It would be helpful a point by point response to the reviewers comments. The ms is still confusing, in the present form tables 2 and 5 of the first version have been combined  and Tables 3 to 5 are lacking, this needs to be revised

Author Response

A point-by-point response was provided during the original resubmission and to the editor (see attached).  The table numbers have been corrected.

Round 2

Reviewer 3 Report

The ms is much clearer in its present form

This manuscript is a resubmission of an earlier submission. The following is a list of the peer review reports and author responses from that submission.

Round 1

Reviewer 1 Report

The presence of Ebola virus (EBOV) in breast milk and semen of human survivors long after recovery from disease raises the question how viral infectivity present in these body fluids can be most efficiently detected. Logue and colleagues compared the efficiency of EBOV, Makona variant, isolation using Vero and Huh-7 cells, monocyte-derived macrophages (MDM) and suckling laboratory mice. Collectively, they find that sensitivity of isolation is comparable between cell lines, MDM and mice and that Huh-7 are slightly more resistant to the cytotoxic effects associated with breast milk as compared to Vero and MDM. This study constitutes, conceptually, a minor step forward but is technically largely solid and should be of some interest to the field. However, several questions remain to be addressed.

Major

EBOV acquires mutations upon passaging in cell lines. For instance, an exchange at position 544 was reported to increase entry into certain human cell lines. Did virus used for the present study acquire mutations reflecting cell culture adaptation during the 3 passages in Vero cells? If so, did they impact isolation efficiency in a cell line dependent manner?

It seems that cytotoxicity of semen and breast milk for cell lines and MDM was determined solely upon microscopic inspection. This approach lacks sensitivity and testing cytotoxicity via a quantitative standardized assay (for instance cell titer glow) is strongly recommended.

This reviewer is surprised that the authors did not test whether mixing virus with breast milk or semen impacts efficiency of EBOV isolation and maybe does so in a cell line dependent fashion. Based on the results obtained with other viruses this is actually quite likely. Since the central goal of this study was to investigate which cell or animal system is most suitable for virus isolation from bodily fluids, these studies should be conducted and the results shown.

Minor

“cell culture-grown EBOV/Mak was serial diluted to a target titer from 0.1 to 10,000 PFU/mL” On which cells was the titer determined?

“ detected in survivors from semen in breast milk long after disease recovery.” This sentence seems grammatically incorrect and should be revised.

Reviewer 2 Report

Ebola virus isolation using Huh-7 cells has methodological advantages and similar sensitivity to isolation using other cell types and suckling BALB/c laboratory mice

In this manuscript, Logue et al evaluated the efficacy of isolating Ebola virus Makona (EBOV/Mak) from three cell lines Vero E6, Huh-7, and monocyte-derived macrophages, and suckling Balb/C laboratory mice. Additionally, the authors compared the effect of alternative matrices, semen and breast milk on toxicity across the three cells lines as well as the suckling mice.  The authors found that a target dose of 10 PFU/ml EBOV/Mak could be successfully isolated from all three cell lines and suckling mice.   Additionally, both Huh-7 cells and suckling mice showed resistance to both breast milk and semen matrix toxicity. 

The major comments:

1.      One of the major comments is on the presentation of the results.  There are too many tables with redundant, overlapping information and not informative columns.  It would be better for the overview of the all isolation results if tables 2, 3 , 4 and 5 can be summarized and simplify in that single or maximum two tables  removing redundant and repetitive information:

a.       For instance, table 3 has information from table 2 about results in cells. It has extra columns like:  “target dose and actual dose” – just use actual dose; “virus isolation form brain and titer” – just put titer as the  most important indication that the isolation was successful;  and so on. 

2.      Other examples of the problems with existing presentation of the data:

a.       In Table 4, the column for “non-surviving laboratory mice” the superscript “b” and explaining text is confusing as the legend explains that “Values were obtained on day 28 post-exposure”  but the mice did not survive to day 28.  This should be reworded for clarification.

b.      In Table 5, the meaning and significance of the numbers in the third row is not clear.

c.       Lines 273-279 refer to “attempts 4 and 5” and later mentioning total of 8 attempts. All that very important information need to be concise in summary table, clearly state how many attempts were made for each cell line, mouse experiment, and what is the titer of successful purifications.

3.      Section 3.5 is summarizing histological findings. It would be very important to have new table summarizing important histological findings including a clear indication of how many mice contained the each lesion (beginning on line 284- end of line 325) and which treatments did not have histological evaluation.  The current technical tone of the text obscures the significance of the findings. 

-          Additionally, it is important to show the histological sections from the saline (or media) control samples and to point the difference in cellular morphology on HE sections.

4.      The section 3.6., about cytotoxicity of breast milk and semen:  In spite of the title authors do not describe an assay to determine and quantitate cytotoxicity. Inspected with microscope cell detachment is merely suggestive of toxicity.  Table 6 need to have quantitative values for each sample.

5.      Finally, the reason of evaluating the toxicity of milk and semen is to be able to bring up and  scale up the viral isolates from those sample.  The authors fail to address the most important questions and show the effect that addition of serum or breast milk matrices has on the ability of cells and mice to be infected by EBOV/Mak and produce virus. Both biological fluids have a lot of proteins, immunological cells and other biologically reactive components and a potential contaminations with bacteria, fungus and mycoplasma that represent a serious problem that can alter infectivity  or even completely obliterate infectivity in vitro models. This concern can be address by adding virus samples at identified concentration to milk or semen and evaluate the ability of discussed procedures to recover and isolate viruses.   

 Minor suggestions:

1.      The manuscript need to be reviewed for the clarity and jargonisms.

2.      The specification of Zika (line 34) in the last phrase of the abstract is not useful and could be even  misleading  because Zika belongs to a different family of viruses with very  different tropism and efficiency of infection in both in vivo and in vitro models.

3.      Material and methods:

a.       Part Virus- the BSL4 mentioned for work with Makona virus and not Mayinga.

b.      Plaque assay –lines 110-113 need to be re-phrased.

4.      In Figure 1 the survival outcome should be included in the graph for clarification of the final mouse number included in the analysis.

5.      For all tables with information of the virus detection it would be beneficial to reduce the amount of abbreviation to essential (for instance in all cases where the assay did not detect virus just put BDL (below detection limit) instead of “ND”, “U”, “-“,  (below the limits of quantification), etc. It is not accurate to provide values and indicate that they are blow detection level.

Reviewer 3 Report

This manuscript by Logue at al. describes the comparison of several cell culture strategies for Ebola virus recovery including in vivo experiments with mice inoculation. In the opinion of the authors the Huh-7 cell line exhibits certain advantages, mainly less toxic effect from samples such as milk or semen, over the standard VeroE6 cells or Monocyte-derived macrophages and mice.

The work to optimize the isolation procedures for highly pathogenic agents such as Ebola virus is extremely important since is obviously restricted to a few BSL4 laboratories than can handle the full infectious agent. In this context, the manuscript contains interesting and valuable information related to practical laboratory procedures to isolate Ebola viruses, however the main conclusion, Huh7 cells are more resistant to matrix toxicity, is not fully supported by the data presented and the whole information contained in the paper needs to be re-organized for clarification.     

Specific comments:

Title is long and rather confusing. Since no advantage is clearly evident a more descriptive title (Comparison of …) would be more appropriated.

There are numerous errors in the composition: (Line 23: viral RNA has been 22 detected in survivors from semen in breast milk long after disease recovery). Line 173: The term EBOV/Yam-Mak is not correct

It is not evident how the present approach would benefit the laboratory procedures for other agents with different cell tropism such as Zika. This comment should be revised (Line 33: parameters tested in this study should be considered when optimizing sample isolation methods for other emerging viruses, such as Zika virus (also repeated in Lines 69-71)

Line 44: Transmission of EVD from male survivors to sexual partners (Ebola virus transmission would be more appropriated) 

There are needless repetitions: Full description of cell lines is also repeated (Line 87, Line 95)

It is very difficult to interpret the tables and figures since complete information is not provided:

Figure 1: Except for the control (n=2) It is not clear the actual number of animal using for each viral dose

Table 2: It is not clear the meaning of Target PFU, Actual PFU and Actual PFU/6 wells. If the Actual PFU/6 wells is just the result of multiplying by 6 is not correct for all values.

Table 4: What is adding two columns for Target dose/mL and Target dose/10uL? How was Actual dose calculated?

Table 5: It would be easier to interpret in a graph.

The main conclusions of the study is that Huh7 are more resistant than Vero or MDM to the cytotoxicity of samples such as milk or semen for Ebola virus isolation is just supported by data in Table 6 from just one well and just for non-clarified milk. These experiments do not require infectious viruses so a more exhaustive approach would be needed to stablish whether Huh7 are more suitable for this type of samples. Also for the sensitivity comparison of the kinetics of growing of Ebola virus in the different cell types would have been valuable.

The manuscript contains potentially relevant information such as the comparison of the pathogenicity of Ebola/Mak vs Yam-May in mice (Figure 1) or the pathology analysis of infected tissues in mice (Figures 2, 3 and 4), but there are nor relevant for the main focus of the study.